# SciReplicate-Bench: Benchmarking LLMs in Agent-driven Algorithmic Reproduction from Research Papers

**Yanzheng Xiang[1], Hanqi Yan[1], Shuyin Ouyang[1], Lin Gui[1], Yulan He[1,2]**
[1]King's College London, [2]The Alan Turing Institute
{yanzheng.xiang,hanqi.yan,shuyin.ouyang,lin.1.gui,yulan.he}@kcl.ac.uk

## Abstract

This study evaluates large language models (LLMs) in generating code from algorithm descriptions in recent NLP papers. The task requires two key competencies: (1) algorithm comprehension: synthesizing information from papers and academic literature to understand implementation logic, and (2) coding expertise: identifying dependencies and correctly implementing necessary APIs. To facilitate rigorous evaluation, we introduce **SciReplicate-Bench**, a benchmark of 100 tasks from 36 NLP papers published in 2024, featuring detailed annotations and comprehensive test cases. Building on SciReplicate-Bench, we propose **Sci-Reproducer**, a dual-agent framework consisting of a Paper Agent that interprets algorithmic concepts from literature and a Code Agent that retrieves dependencies from repositories and implements solutions. To assess algorithm understanding, we introduce *reasoning graph accuracy*, which quantifies similarity between generated and reference reasoning graphs derived from code comments and structure. For evaluating implementation quality, we employ *execution accuracy*, *CodeBLEU*, and repository dependency/API *recall* metrics. In our experiments, we evaluate various powerful non-reasoning and reasoning LLMs as foundational models. The best-performing LLM using Sci-Reproducer achieves only 39% *execution accuracy*, highlighting the benchmark's difficulty. Our analysis identifies missing or inconsistent algorithm descriptions as key barriers to successful reproduction. We make available our benchmark and code at ⭕ GitHub and project homepage at 🔗 Homepage.

## 1 Introduction

The evolution of Large Language Models (LLMs) has ushered in a transformative era in scientific discovery, positioning them as powerful tools for streamlining research (Gridach et al., 2025; Buehler, 2024; Lu et al., 2024), from idea generation to verification and publication writing. For instance, Si et al. (2025); Gu & Krenn (2025) demonstrated how LLMs can be prompted to generate novel research ideas, while Yuan et al. (2022); Du et al. (2024) explored their use in producing literature reviews for idea evaluation. Additionally, LLMs are increasingly integrated into tools like Semantic Scholar[1], Research Rabbit[2], and Undermind Research Assistant[3], enhancing literature discovery, citation analysis, and knowledge synthesis. These advancements, both in research methodologies and practical applications, suggest that LLMs have the potential to assist across multiple stages of scientific discovery.

Among the aforementioned advancements in research acceleration, the ability of LLMs to correctly generate code for validating real-world scientific ideas is particularly noteworthy. Computational validation is crucial across many fields, yet researchers often face barriers due to limited coding expertise or inaccessible implementations. By converting scientific

---

[1]https://www.semanticscholar.org
[2]https://www.researchrabbit.ai/
[3]https://www.undermind.ai/

algorithm descriptions into executable code, LLMs could enhance reproducibility and accelerate scientific discovery. However, despite progress in LLM-based code generation, a significant gap remains in generating code directly from scholarly papers. ***First***, *algorithm comprehension from scientific papers is challenging*. Research papers are characterized by their brevity, methodological rigor, and extensive citations, with critical details about algorithms often dispersed across multiple sections of the paper. Understanding these algorithms requires synthesizing information from internal references and external scholarly works. ***Second***, *code repositories typically consist of multiple interdependent files and directories*. To implement an algorithm, LLMs must comprehensively examine file dependencies, identify reusable components, and correctly handle both internal dependencies and external APIs.

Despite the importance of automated scientific idea verification, there exists no dataset specifically designed to evaluate the ability of LLMs to reproduce real-world algorithms proposed in peer-reviewed publications. As shown in Table 1, there are several machine learning software engineering benchmarks primarily focused on evaluating algorithmic design or straightforward implementations, which are significantly less complex than the methods typically described in academic research papers. For example, MLE-BENCH (Liu et al., 2023) and MLAgentBench (Huang et al., 2023) utilize Kaggle competitions, where LLMs must develop and implement solutions based on provided task specifications. ML-BENCH (Chan et al., 2024) uses Machine Learning (ML) GitHub repositories to assess LLMs' text-to-code capabilities and test autonomous agents in task execution.

| Benchmark | Paper Understanding | Repo-Search | Test Case | Source | Task Types |
|---|:---:|:---:|:---:|---|---|
| MLE-BENCH | ✗ | ✗ | ✗ | Kaggle | Algorithm design and code gen. |
| MLAgentBench | ✗ | ✓ | ✓ | Kaggle | Algorithm design and code gen. |
| ML-BENCH | ✗ | ✓ | ✓ | Github | Code gen. |
| SciReplicate-Bench | ✓ | ✓ | ✓ | Publications | Replicate code for algorithms in real-world NLP publications. |

Table 1: Comparisons of different machine learning software engineering benchmarks.

To address this gap, we developed **SciReplicate-Bench, the first benchmark specifically designed to evaluate LLMs' capabilities in code generation for reproducing research findings from academic papers**. It consists of 100 code reproduction tasks derived from 36 papers published in leading conferences in 2024. This recent publication window was deliberately chosen to minimize the risk of data leakage. An overview of the task is illustrated in Figure 1, with a concrete example provided in Figure A2 in Appendix E. The task consists of two main steps: **1. Algorithm understanding**. LLMs must extract essential information from the paper, such as workflow details, algorithm descriptions, and hyperparameter values. **2. Code implementation**. LLMs then implement a function or method within a provided repository, using both the extracted information and the LaTeX representation of the algorithm from the paper. We introduce **Sci-Reproducer**, a dual-agent system that combines a Paper Agent and a Code Agent to handle these two steps collaboratively and implement code for the target algorithm.

To rigorously assess LLM performance on this benchmark, we evaluate two dimensions corresponding to the aforementioned two steps: *algorithm comprehension correctness* and *code correctness*. To evaluate algorithm comprehension, we introduce a **reasoning graph** to represent the reasoning logic behind algorithm reproduction. Each node in the graph represents a code comment, which reflects a single reasoning step and is aligned with a specific segment of code. Edges between nodes are defined based on data flow relationships across different code segments. We compute the similarity between the generated reasoning graph and a reference graph to derive the *reasoning graph accuracy*. To evaluate code correctness, we employ established metrics including *execution accuracy* (Rajkumar et al., 2022; Xiang et al., 2023), *CodeBLEU* (Ren et al., 2020), and *recall* of intra/cross-file dependencies and APIs.

Our work makes the following contributions:

**Benchmarks**: SciReplicate-Bench, a benchmark of 100 algorithm reproduction tasks from recent NLP publications.

**Metric**: We propose a novel *reasoning graph accuracy* metric for evaluating algorithmic comprehension.

**Approach**: Sci-Reproducer, a dual-agent framework combining paper understanding and code implementation.

**Insights**: Comprehensive evaluation across state-of-the-art LLMs reveals four key findings: (i) the task remains highly challenging, with execution accuracy below 40% for all models; (ii) reasoning models exhibit "overthinking" behavior (Cuadron et al., 2025; Sui et al., 2025), over-relying on internal reasoning rather than utilizing available tools for information extraction; (iii) while LLMs demonstrate strong algorithmic comprehension, they struggle with practical implementation; and (iv) algorithm reproduction failures often stem from incomplete paper descriptions, which our Sci-Reproducer effectively addresses.

## 2 Related Work

Our work lies at the intersection of AI for automating scientific discovery and LLM-based code generation.

### 2.1 AI for Automating Scientific Discovery

The application of LLMs to accelerate scientific research has emerged as a rapidly growing field with diverse approaches. Several studies have demonstrated the potential for comprehensive research automation through end-to-end AI systems. Schmidgall et al. (2025); Lu et al. (2024) developed frameworks that integrate idea generation, experimental validation, and manuscript composition, with some AI-authored papers successfully passing workshop review processes (Yamada et al., 2025). Complementary research has focused on the creative aspects of scientific inquiry, with Wang et al. (2023); Ghafarollahi & Buehler (2024); O'Neill et al. (2025) investigating LLMs' capacity for generating novel research hypotheses. Notably, recent evaluations by Gu et al. (2024); Kumar et al. (2024); Liu et al. (2025); Si et al. (2024) suggest that AI-generated research concepts may occasionally exceed human-generated ideas in terms of novelty and originality.

Within computational disciplines where implementation validation is essential, LLMs have shown promise in algorithm design and code development tasks. Our proposed SciReplicate-Bench addresses a previously underexplored area: the automated reproduction of algorithms directly from academic publications. This represents a unique challenge at the intersection of scientific literature comprehension and executable code synthesis. While recent parallel efforts such as PaperBench (Starace et al., 2025) and Paper2CodeBench (Seo et al., 2025) have tackled related problems by exploring full codebase reconstruction, these evaluation approaches rely substantially on manual assessment criteria and LLM-based correctness judgments, introducing potential inconsistencies and reliability concerns. Our approach prioritizes objective evaluation through *execution accuracy*, providing more rigorous validation than non-executable assessment methodologies (Wang et al., 2022; Chen et al., 2021).

The broader ecosystem of computational reproducibility research includes specialized frameworks such as MLGym (Nathani et al., 2025) for baseline improvement, and evaluation benchmarks developed by Siegel et al. (2024); Ren et al. (2023) that assess LLMs' ability to reproduce published experimental results using existing codebases.

### 2.2 LLMs for Code Generation

Code generation has emerged as a prominent application of LLMs, with benchmarks ranging from basic programming tasks (Chen et al., 2021; Jain et al., 2024; Austin et al., 2021; Hendrycks et al., 2021; Liu et al., 2022) to realistic software engineering challenges like SWE-bench (Jimenez et al., 2023), which uses actual repository pull requests. However, these benchmarks primarily target general software engineering rather than scientific algorithm reproduction.

Recent efforts have developed machine learning-specific benchmarks (Liu et al., 2023; Huang et al., 2023; Chan et al., 2024), but these typically involve implementing algorithms proposed by the models themselves or solving relatively straightforward tasks. They lack the depth of algorithmic understanding and rigorous paper analysis required for reproducing algorithms from peer-reviewed publications.

Despite advances in tool-augmented code generation (Schick et al., 2023; Zhang et al., 2024a;a; 2023b), no existing system specifically addresses the unique challenge of translating academic papers into executable code. Our Sci-Reproducer framework demonstrates the ability to comprehend academic publications and convert abstract algorithm descriptions into functional implementations.

## 3 SciReplicate-Bench

**Overview**   SciReplicate-Bench is designed to evaluate LLMs' ability to reproduce algorithms from academic papers, consisting of 100 tasks curated from 36 recent NLP publications with their corresponding open-source implementations. The task categories are detailed in Figure A1 in Appendix B. The benchmark focuses on repository-level code generation, where each task is centered around implementing a specific function or class method. As illustrated in Figure A3 in Appendix E, each task comprises nine components, which can be categorized into three groups corresponding to code generation, evaluation, and analysis, respectively.

For code generation, the following components are provided as inputs to LLMs:

*Function signature*: the definition of the target function, including detailed descriptions of its input and output variables.

*Algorithm Description*: The LaTeX code description of the target algorithm, typically located within a subsection or paragraph of the target paper.

*Literature context*: the original paper along with its cited references, providing broader conceptual context.

*Repository context*: all source files and code in the repository that inform or support the target implementation.

For evaluation, the following components are provided for code execution and metrics calculation:

*Reference implementation*: ground-truth code serving as the reference for *CodeBLEU* evaluation.

*Reasoning graph annotations*: structured representations of the algorithmic logic and implementation flow, enabling assessment of *reasoning graph accuracy*.

*Dependency annotations*: comprehensive documentation of internal dependencies, cross-file relationships, and external API usage for computing *recall* metrics across all dependency categories.

*Test environment*: isolated Python execution environment containing validation cases and automated verification scripts for assessing implementation correctness.

To enable further analysis of the underlying causes of LLM failures, the benchmark includes:

*Missing/Mismatch Information*: the LaTeX description of the algorithm may omit certain implementation details, which could either appear elsewhere in the paper or be entirely absent. We also annotate mismatches between the paper description and the reference implementation.

**Task Definition**   Based on SciReplicate-Bench, an LLM is given the algorithm description, function signature, literature context, and repository context as input. The LLM is asked to output a function that implements the target algorithm.

**Benchmark Construction**   The benchmark construction process comprises four key steps: paper selection, Python environment setup, documentation, and verification suite preparation. To mitigate the risk of data leakage, we selected papers published in 2024 that

provide publicly available code repositories. During the annotation process, each repository was refactored to isolate the core algorithm into a standalone function, and all sources of randomness were removed to ensure reproducibility and prevent leakage. On average, annotating each paper requires approximately 12 hours. Details of the annotation process are provided in Appendix A.

## 3.1 Evaluation Metrics

### 3.1.1 Evaluating Algorithm Comprehension

We propose the *reasoning graph accuracy* metric to evaluate how well LLMs understand the logic and implementation of algorithms. During code generation, LLMs are prompted to insert specially formatted, non-overlapping, non-nested comments that mark reasoning steps derived from the algorithm's LaTeX code (The prompt can be found in Figure A5). We then construct a reasoning graph $G = \{N, E\}$ (illustrated in Figure A3), modeled as a Directed Acyclic Graph (DAG). Each node $n_i = \langle w_i, c_i \rangle, n_i \in N$ represents a reasoning step with a comment $w_i$ and corresponding code snippet $c_i$. An edge $e_i = \langle n_i, n_j \rangle, e_i \in E$ is added if a variable used in $c_j$ is defined or last modified in $c_i$. To compute the *reasoning graph accuracy*, we compare the generated graph $G_g$ with the reference graph $G_r$ via node and edge matching:

**Node matching:** comments from $G_r$ and $G_g$ are passed to GPT-4o, which maps each reference node to one or more nodes in the generated graph. A node in $G_r$ is considered matched if it has at least one corresponding node in $G_g$. The prompt template used for this process is available in Figure A4.

**Edge matching:** for each reference edge $e_r = \langle n_r^i, n_r^j \rangle$, if both endpoint nodes have corresponding matches in $G_g$, we apply Breadth-First Search(BFS) to verify whether a corresponding edge exists in $G_g$.

The *reasoning graph accuracy* $S_r$ is computed as:

$$S_r = \sum_{n_i}^{n_i \in N_m} s_i^n + \sum_{e_j}^{e_j \in E_m} s_j^e. \tag{1}$$

where $N_m$ and $E_m$ denote the sets of matched nodes and edges, respectively, and $s_i^n$ and $s_j^e$ represent their corresponding significance scores. Node significance is determined by the complexity of its corresponding code segment, measured by the number of variable definitions and usages, function calls, arithmetic operations, and lines of code, then normalized across the reference graph. Edge significance is calculated as the product of the significance scores of its connected nodes, followed by normalization.

### 3.1.2 Evaluating Code Generation

For assessing coding ability, we use the following evaluation metrics:

- *Execution accuracy* (Xiang et al., 2023; Zhang et al., 2024b; Long et al., 2022): we integrate the generated code into the repository and execute it to obtain results. If all test cases match the reference results, we consider the code correct.

- *CodeBLEU* (Ren et al., 2020): this metric evaluates how similar the generated code is to reference code by using the traditional BLEU metric (Papineni et al., 2002) while incorporating syntactic information through abstract syntax trees (AST) and semantic understanding via data-flow graphs (DFG).

- *Recall* (Li et al., 2024): we calculate recall scores specifically for intra-file dependencies, cross-file dependencies, and external APIs.

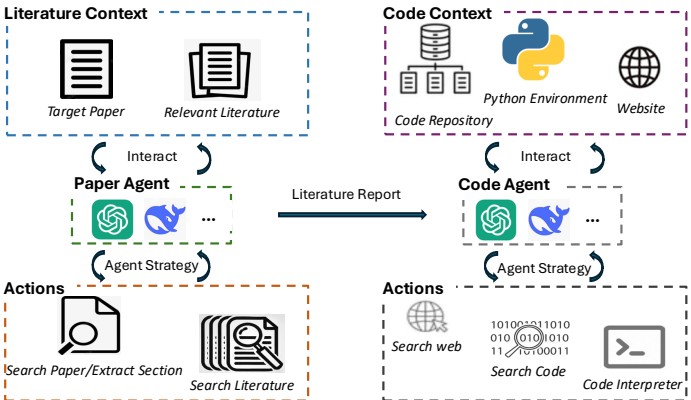

Figure 1: Overview of the task and the proposed Sci-Reproducer framework. The task involves algorithm understanding and code implementation, handled by a Paper Agent and a Code Agent operating in separate contexts with specialized actions.

| Action Name | Input | Observation |
|---|---|---|
| **Paper Agent** | | |
| SearchPaper | Query | The retrieved response from the target paper in relation to the query. |
| SearchSection | Section ID | The entire content of a section based on the section label. |
| SearchLiterature | Paper ID, query | The answer to the query searched from the literature (identified by Paper ID). |
| **Code Agent** | | |
| SearchCode | Name | The definition of a specific code element in repository. |
| SearchFile | Name | The content of a certain file in repository. |
| SearchWeb | Query | The information obtained from the website. |
| Compiler | code | The feedback from the compiler after executing the code. |

Table 2: The pre-defined actions for the Paper Agent and the Code Agent.

# 4 Sci-Reproducer

To address this task, we introduce Sci-Reproducer [4], a dual-agent framework designed for scientific paper algorithm replication. As illustrated in Figure 1, Sci-Reproducer comprises a Paper Agent and a Code Agent that collaboratively work to replicate algorithms described in a given paper. The predefined actions employed by the agents are summarized in Table 2, with implementation details provided in Appendix C.

## 4.1 Paper Agent

Based on the provided algorithm description, the Paper Agent systematically retrieves contextual information from the literature context to support algorithmic understanding. Due to the input length limitations of LLMs, it is infeasible to input entire paper along with their associated literature. Consequently, the Paper Agent must selectively extract pertinent information, following a strategy akin to Retrieval Augmented Generation (RAG) (Wang et al., 2024b; Sarthi et al., 2024). The Paper Agent incrementally builds an understanding of the target algorithm by executing predefined actions to query the literature context. To facilitate this process, we adopt ReAct (Yao et al., 2022) as the agent strategy, which enables seamless integration of action execution with intermediate reasoning steps.

After the Paper Agent concludes that all necessary information has been collected, it generates a comprehensive literature report comprising key findings that fill in the missing

---

[4]A video demonstration showcasing Sci-Reproducer's capabilities is provided at https://youtu.be/qcSIMgyehjE

| Approach | Exe Acc(↑) | CodeBLEU(↑) | RG Acc(↑) | Recall Intra-File(↑) | Cross-File(↑) | API(↑) |
|---|---|---|---|---|---|---|
| 🌀 **GPT-4o-mini** | | | | | | |
| No Agent | 0.030 | 0.238 | 0.694 | 0.012 | 0.000 | 0.217 |
| Paper Agent | 0.040 | 0.246 | 0.717 | 0.024 | 0.000 | 0.251 |
| Code Agent | 0.140 | 0.279 | 0.738 | 0.565 | 0.364 | 0.328 |
| Sci-Reproducer | **0.170** | **0.303** | **0.741** | **0.576** | **0.364** | **0.362** |
| 🌀 **GPT-4o** | | | | | | |
| No Agent | 0.040 | 0.259 | 0.705 | 0.059 | 0.000 | 0.281 |
| Paper Agent | 0.020 | 0.263 | 0.724 | 0.023 | 0.000 | 0.298 |
| Code Agent | 0.260 | 0.325 | 0.748 | 0.682 | 0.576 | **0.421** |
| Sci-Reproducer | **0.270** | **0.329** | **0.751** | **0.688** | **0.636** | 0.417 |
| ⓐ **Claude-Sonnet-3.7** | | | | | | |
| No Agent | 0.070 | 0.282 | 0.725 | 0.094 | 0.091 | 0.362 |
| Paper Agent | 0.050 | 0.291 | 0.727 | 0.082 | 0.091 | 0.379 |
| Code Agent | 0.320 | 0.394 | 0.764 | 0.765 | 0.545 | 0.545 |
| Sci-Reproducer | **0.390** | **0.401** | **0.773** | **0.776** | **0.636** | **0.626** |
| ✦ **Gemini-2.0-Flash** | | | | | | |
| No Agent | 0.070 | 0.275 | 0.688 | 0.071 | 0.000 | 0.294 |
| Paper Agent | 0.040 | 0.278 | 0.699 | 0.082 | 0.000 | 0.332 |
| Code Agent | 0.220 | 0.323 | 0.708 | 0.553 | 0.212 | 0.426 |
| Sci-Reproducer | **0.250** | **0.346** | **0.727** | **0.588** | **0.333** | **0.455** |
| 🐋 **Deepseek-V3** | | | | | | |
| No Agent | 0.030 | 0.260 | 0.712 | 0.012 | 0.061 | 0.272 |
| Paper Agent | 0.050 | 0.275 | 0.732 | 0.012 | 0.030 | 0.306 |
| Code Agent | 0.210 | 0.312 | 0.736 | 0.482 | 0.182 | 0.383 |
| Sci-Reproducer | **0.220** | **0.334** | **0.738** | **0.565** | **0.333** | **0.443** |
| 🌀 **o3-mini-low** 💬 | | | | | | |
| No Agent | 0.080 | 0.259 | 0.758 | 0.035 | 0.000 | 0.323 |
| Paper Agent | 0.050 | 0.262 | 0.729 | 0.035 | 0.000 | 0.315 |
| Code Agent | 0.150 | 0.278 | **0.803** | 0.306 | 0.000 | **0.348** |
| Sci-Reproducer | **0.180** | **0.280** | 0.771 | **0.376** | **0.121** | 0.328 |
| 🌀 **o3-mini-medium** 💬 | | | | | | |
| No Agent | 0.040 | 0.263 | 0.729 | 0.035 | 0.000 | 0.336 |
| Paper Agent | 0.060 | 0.263 | 0.726 | 0.047 | 0.000 | 0.319 |
| Code Agent | 0.220 | **0.289** | 0.749 | **0.376** | 0.030 | **0.404** |
| Sci-Reproducer | **0.240** | 0.283 | **0.758** | 0.341 | **0.061** | 0.362 |
| 🌀 **o3-mini-high** 💬 | | | | | | |
| No Agent | 0.070 | 0.269 | 0.718 | 0.047 | 0.000 | 0.345 |
| Paper Agent | 0.070 | 0.267 | 0.751 | 0.035 | 0.000 | 0.366 |
| Code Agent | 0.160 | 0.277 | **0.773** | 0.165 | **0.152** | **0.374** |
| Sci-Reproducer | **0.160** | **0.283** | 0.763 | **0.294** | 0.091 | 0.357 |

Table 3: Performance evaluation on the SciReplicate-Bench benchmark. Models with 💬 notation indicate **reasoning LLMs**. "Exe Acc" represents *execution accuracy* while "RG Acc" indicates *reasoning graph accuracy*.

components of the target algorithm's LaTeX source. An example of the literature report is shown in Figure A8. This report subsequently serves as a crucial input for the Code Agent. The prompt used to guide the Paper Agent is provided in Figure A6.

### 4.2 Code Agent

Informed by the algorithm description, literature report, and code context, the Code Agent searches the repository to locate essential dependencies required for implementation. It can also browse websites for additional information and use a compiler to test and iteratively debug the code, ensuring proper execution by identifying and fixing syntax errors. The prompt for the Code Agent is provided in Figure A7.

## 5 Experiments

We evaluate Sci-Reproducer on the SciReplicate-Bench benchmark using 7 advanced LLMs, including five non-reasoning LLMs: GPT-4o-mini (4o mini, 2024), GPT-4o (GPT-4o, 2024), Claude-Sonnet-3.7 (Claude-Sonnet-3.7, 2025), Gemini-2.0-Flash (Gemini-2.0-Flash, 2024),

and Deepseek-V3 (DeepSeek-AI et al., 2024), and different versions of the reasoning models O3-mini (o3 mini, 2024), i.e., three different levels of reasoning intensity. For the *reasoning graph accuracy* metric, node matching is performed using GPT-4o, which may introduce some randomness. To reduce this variability, we set the temperature to 0 and top-p to 1, ensuring more deterministic generation. The calculation is repeated three times, and we report the average score as the final result.

## 5.1 Results on SciReplicate-Bench

Table 3 displays Sci-Reproducer's evaluation results and contributions of Code/Paper Agent. The "No Agent"directly prompts the LLM to generate code based solely on the algorithm description and function signature. "No Paper Agent" allows the LLM to use Code Agent actions, but restricts access to Paper Agent actions. "No Code Agent" grants access to Paper Agent actions but blocks Code Agent capabilities. The results offer key insights, discussed in the following.

**LLMs struggles on SciReplicate-Bench** Most LLMs perform poorly, achieving less than 0.1 *execution accuracy* without using the agent to examine literature and repository contexts. With enhancement of Sci-Reproducer, these LLMs show notable improvements, with an average increase of 0.181 in execution ACC and 0.057 in *CodeBLEU*, although even the best-performing model, Claude-Sonnet-3.7, only achieved 0.390 *execution accuracy*. This highlights the exceptional challenge presented by our SciReplicate-Bench.

**LLMs can comprehend algorithm logic** LLMs demonstrate strong algorithmic comprehension capabilities, as evidenced by *reasoning graph accuracy scores* averaging 0.716 even without agent assistance. The addition of individual agents provides modest but consistent improvements: the Paper Agent increases understanding by 0.009 on average, while the Code Agent contributes a larger gain of 0.036. Combined agent deployment yields a cumulative improvement of 0.037. These enhancements stem from complementary mechanisms: the Paper Agent strengthens theoretical comprehension by gathering relevant contextual information from academic literature, while the Code Agent facilitates practical understanding through extraction of pertinent code patterns and dependency structures from repositories.

**LLMs face challenges with actual implementation** Although LLMs are capable of understanding algorithms, their performance in code generation remains suboptimal. Despite using Sci-Reproducer, the average *execution accuracy* remains low at 0.235, with a *CodeBLEU* score of 0.320.

**Accurate dependency and API identification is crucial for code implementation** Effectively recognizing and leveraging dependencies from the source repository and external APIs is essential for accurate code implementation. The integration of Code Agent led to substantial gains in *recall* with average increases of 0.441, 0.239, and 0.100, respectively, compared to cases without the agent. With Sci-Reproducer, Claude-Sonnet-3.7 attains the highest *execution accuracy* of 0.390, with the highest *recall* for intra/cross file dependency and API usage, at 0.776, 0.636, and 0.626 respectively.

**Overthinking leads to limited improvement in reasoning LLMs** Reasoning LLMs exhibit more modest performance gains when using Sci-Reproducer. While they achieve an average *execution accuracy* improvement of 0.13, non-reasoning models demonstrate substantially larger gains of 0.212. This pattern extends to *recall* metrics, where reasoning LLMs show improvements of 0.243, 0.061, and 0.041 respectively, compared to non-reasoning LLMs' more pronounced gains of 0.560, 0.345, and 0.135. We attribute this performance gap to the "overthinking" phenomenon (Cuadron et al., 2025; Sui et al., 2025), where excessive internal reasoning impedes effective action execution, a limitation that we examine in detail in the following subsection.

## 5.2 Tool Usage Analysis

Figure 2 presents the number of times each LLM invokes actions on the full dataset with Sci-Reproducer. We observe the following:

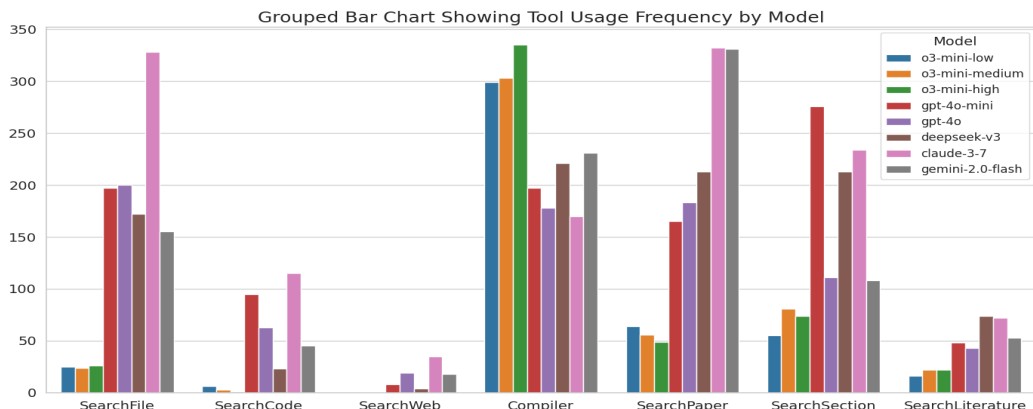

Figure 2: A grouped bar chart illustrating the frequency of tool usage by different models. The x-axis represents various actions, while the y-axis indicates the total number of times each tool was used on this dataset.

- For code-related actions, reasoning LLMs demonstrate limited tool usage, employing "SearchFile", "SearchCodeItem", and "SearchWeb" only 25.0, 3.3, and 0.0 times on average, respectively. Non-reasoning LLMs use these same actions far more extensively, with averages of 210.4, 68.2, and 16.8 times respectively. This disparity reveals a fundamental behavioral difference: reasoning models favor internal deliberation over external information gathering. Conversely, reasoning LLMs invoke "Compiler" more frequently, indicating they require more debugging iterations due to inadequate contextual information gathering. This over-reliance on internal reasoning undermines performance: advanced models like o3-mini-high and o3-mini-low achieve *execution accuracy* comparable to GPT-4o-mini, negating their theoretical computational advantages.

- Paper-related actions exhibit a similar pattern. Reasoning LLMs use "SearchPaper", "SearchSection", and "SearchLiterature" an average of 56.3, 70.0, and 20.0 times respectively, while non-reasoning LLMs demonstrate substantially higher usage at 244.8, 188.4, and 58.0 times respectively. Additionally, we observe a clear preference for target paper extraction over external literature consultation. Actions targeting the primary paper ("SearchPaper" and "SearchSection") are invoked 174.1 and 144 times on average, significantly more than "SearchLiterature" which accesses related works only 43.8 times.

### 5.3 Error Analysis

#### 5.3.1 Syntax Errors

Table A4 shows the syntax error rates for each model across different configurations. Without the Code Agent, syntax errors occurred at rates of 80.3% ("NoAgent") and 83.3% ("Paper Agent"). After implementing the Code Agent, these error rates dropped significantly to 29.4% ("Code Agent") and 24.9% ("Sci-Reproducer"). The remaining syntax errors mainly result from incorrectly using repository dependencies. This occurs because our approach, unlike human developers, cannot dynamically access runtime information through a compiler during the code generation process.

#### 5.3.2 Logic Errors

Another issue stems from differences in implementation logic, which can be broadly categorized into: (1) discrepancy in algorithm implementation that result in differing outputs, and (2) missing or mismatch information in the algorithm descriptions in the paper compared to the actual code.

**Implementation discrepancy** An algorithm may have multiple valid implementation approaches. For example, the cross-entropy loss function can be implemented by directly

| Model (Sci-Reproducer) | Exe Acc(↑) | CodeBLEU(↑) | RG Acc(↑) | Recall | | API(↑) |
| | | | | Intra-File(↑) | Cross-File(↑) | |
|---|---|---|---|---|---|---|
| 🌀 GPT-4o-mini | 0.220 | 0.316 | 0.809 | 0.588 | 0.485 | 0.409 |
| 🐋 Deepseek-V3 | 0.470 | 0.378 | 0.834 | 0.682 | 0.424 | 0.609 |
| 🌀 o3-mini-low 🐱 | 0.220 | 0.292 | 0.850 | 0.259 | 0.091 | 0.460 |

Table 4: Experimental Results when missing/mismatched information is regard as external input in the prompt.

invoking the PyTorch API "torch.nn.CrossEntropy" or by manually coding it from scratch. Such implementation choices may introduce subtle differences that lead to variations in the final output of the function.

**Missing/Mismatched information in algorithm description**  Algorithmic descriptions in research papers often lack concrete implementation details, and in certain cases, the provided code may exhibit minor discrepancies compared to the descriptions in the paper. We manually compared the implementation code of all tasks in the dataset with their descriptions in the papers to identify missing or mismatch information. We then provided this information as additional input and apply Sci-Reproducer framework on three LLMs. The Results is shown in Table 4, regarding to Execution Acc, the performance for GPT-4o-mini, Deepseek-V3 and O3-mini-low improved 0.050, 0.250 and 0.040 respectively. The missing information can be divided into four categories:

- Hyperparameters and configurations: descriptions of target algorithms in papers often omit specific hyperparameter settings, such as the batch size.
- Numerical stability techniques: standard techniques for ensuring numerical stability, such as handling division by zero.
- Implementation logic: common implementation practices and model design choices, such as data splitting protocols.
- Coding strategy: practical programming techniques that enhance implementation efficiency and reliability, such as early stopping criteria.

More examples for each category can be found in Table A5 in Appendix E. As for mismatched information, it occurs far less frequently compared to missing information, and its categories largely overlap with those mentioned above.

To mitigate the widespread issues of missing and mismatched information, the first category can generally be addressed by referencing the original research paper and related literature, or by inspecting the code repository for explicit configurations. However, addressing the other three categories requires familiarity with general machine learning coding conventions, thus necessitating that the LLMs identify and utilize implementation patterns from comparable algorithms to enhance code quality. Future research may improve performance by incorporating implementation insights from similar algorithms through techniques such as in-context learning (Zhou et al., 2023; Xiang et al., 2024), and by leveraging real-time compiler feedback to infer precise variable values.

## 6  Conclusion

We evaluate LLMs' ability to replicate algorithms described in recent NLP papers. To support this, we introduce SciReplicate-Bench, a benchmark with rich annotations, and Sci-Reproducer, a dual-agent framework for bridging algorithm understanding and code generation. We assess performance using *reasoning graph accuracy* and standard implementation metrics. Results show the task is highly challenging, with failures largely caused by missing or inconsistent algorithm descriptions.

## 7  Acknowledgements

This work was supported in part by the UK Engineering and Physical Sciences Research Council (EPSRC) through a Turing AI Fellowship (grant no. EP/V020579/1, EP/V020579/2). We thank the authors of the selected papers for making their code openly available.

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

## Appendix

## A    Details of the Annotation Process

**Step 1: paper selection**    We curated NLP papers from leading conferences in 2024, including ACL, EMNLP, ICLR, Neurips, and COLING. Using a web crawler, we collected accepted paper titles and employed the PapersWithCode API[5] to identify those with open-source implementations. For each identified paper, we retrieved corresponding GitHub repository links and metadata (e.g., stars, issues, release dates) via the GitHub REST API[6].

To filter candidates, we applied the following criteria:

- Removed survey/exploratory papers while retaining method-focused research.
- Applied a cutoff date of January 1, 2024 to avoid data leakage.
- Excluded repositories with fewer than 5 stars to ensure basic quality assurance.

Subsequently, researchers manually reviewed each candidate paper and its repository. We discarded papers with excessive computational demands, poorly structured code, ambiguous documentation, missing preprocessing steps, or reported reproduction issues.

**Step2: python environment setup**    For papers passing the initial screening, annotators followed the README to set up the environment and replicate experiments. Common issues included dependency conflicts, data loading failures, and incomplete or buggy code. Annotators attempted to resolve these problems; repositories with irrecoverable errors were excluded.

**Step3: annotation**    Annotation consists of two steps:

1. Algorithm-Function alignment: most papers contain multiple algorithmic components, often organized as subsections. Annotators segmented these into distinct units and mapped each to its corresponding implementation. Code was refactored to encapsulate each algorithm in a standalone function or method. Papers with implementations too fragmented for restructuring were excluded.

---

[5]https://paperswithcode.com/api/v1/docs/
[6]https://docs.github.com/en/rest?apiVersion=2022-11-28

2. Detailed annotation: for each aligned function, annotators documented input/output variables, intra- and cross-file dependencies, and external API usage. Additionally, they inserted explanatory comments mapping code segments to algorithm components. Based on these annotations and variable dependencies, we can construct a reasoning graph representing the implementation logic. During the annotation process, LLMs were employed to assist with algorithm-function alignment and the generation of variable descriptions and code comments. All outputs were subsequently reviewed and corrected by human annotators to ensure accuracy.

The final selected papers are listed in Table A1.

**Step 4: verification suite preparation**  Finally, annotators created verification suites with 10 test cases per task, drawn from the original datasets used in each repository for the majority of papers. For a small number of repositories, fewer than 10 test cases could be constructed. For instance, algorithms that analyze LLM parameters may have only a single test case. Given the inherent randomness in many NLP implementations and potential machine-related variability, we addressed reproducibility from two angles:

- Eliminating code randomness: annotators fixed random seeds and replaced nondeterministic operations (e.g., unordered sets) with deterministic equivalents to ensure consistent outputs across runs.
- Controlling hardware variability: users were instructed to run both reference and generated code locally to eliminate discrepancies caused by hardware differences.

Lastly, annotators implemented task-specific comparison scripts to evaluate output correctness, accounting for variations in return types across tasks.

## B  Details of the Task Categories

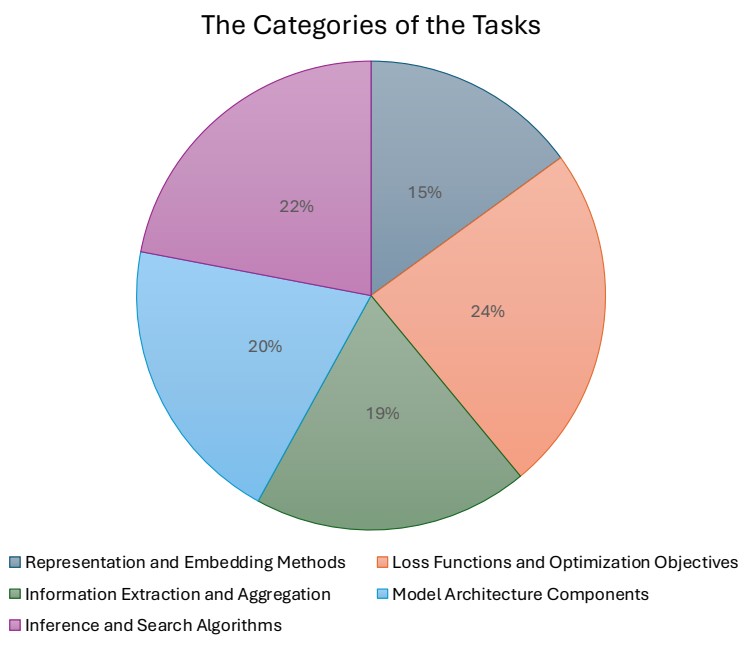

Figure A1: The categories of the tasks within SciReplicate-Bench.

The benchmark encompasses five main task categories in the NLP domain: representation and embedding methods, loss functions and optimization objectives, information extraction and aggregation, model architecture components, and inference and search algorithms. The distribution of each task category is illustrated in Figure A1.

## C   Details of the Actions

In this section, we provide implement details for all actions defined in the Sci-Reproducer.

**SearchPaper**   We obtain the LaTeX source code of the target academic paper from arXiv[7] and apply regular expression-based parsing to extract the content corresponding to each section. Subsequently, we iteratively feed the content of each subsection, along with the query generated by the large language model, into GPT-4o-mini. The model extracts relevant information and returns it as an observation to the paper agent.

**SearchSection**   Following the same approach as `SearchPaper`, the tool begins by parsing the LaTeX source code of the target algorithm. Upon receiving a section ID from the Paper Agent, it retrieves and returns the content of the corresponding section.

**SearchLiterature**   Given a paper ID and a query, the tool attempts to download the corresponding LaTeX source code from arXiv. If the LaTeX source code is unavailable, it returns no information. Otherwise, it extracts content relevant to the query from the paper, following the same procedure as the `SearchPaper` action.

**SearchCode**   For each Python file in the code repository, we utilize the Python AST [8] package to parse the file and extract all defined classes, functions, and global variables. Unlike embedding-based code search methods (Zhang et al., 2024c; 2023c), the Code Agent in our framework directly provides the name of a code item. The tool then returns the corresponding definition if it exists; otherwise, it returns an empty response.

**SearchFile**   When the Code Agent provides a file name, the tool returns the full content of the corresponding file.

**SearchWeb**   When the Code Agent issues a query, we use the Google Search API [9] to retrieve relevant information from websites. These results are then processed by GPT-4o-mini, which filters the content and extracts the information most relevant to the query for return.

**Compiler**   Once the Code Agent completes code generation, it invokes the compiler to execute the code. The generated function or method is inserted into the original Python file, and the corresponding Python environment is used to run the code. The output from the compiler is then returned as the feedback.

## D   Human Evaluation of Reasoning Graph Accuracy

To validate the reliability of our LLM-based evaluation metrics, we conducted a human evaluation study on a subset of our benchmark. Three PhD students in computer science independently assessed the reasoning graph accuracy for 20 tasks generated by GPT-4o-mini and O3-mini-medium using Sci-Reproducer.

As shown in Table A2, human evaluations demonstrate strong alignment with our automated LLM-based assessments. The mean scores show consistent agreement between human annotators and GPT-4o evaluations, with standard deviations indicating reasonable inter-annotator consistency.

Furthermore, we computed the Pearson correlation coefficient between human and LLM-based evaluations across all assessed instances. As presented in Table A3, the correlation ($r = 0.7518$, $p < 0.005$) indicates a strong positive relationship between human judgments and automated assessments, supporting the validity of our LLM-based evaluation approach.

---

[7] https://arxiv.org/
[8] https://docs.python.org/3/library/ast.html
[9] https://developers.google.com/custom-search/v1/

While this preliminary validation demonstrates the effectiveness of our automated metrics, comprehensive human evaluation across the full benchmark remains an important direction for future work.

# E   Figures and Tables

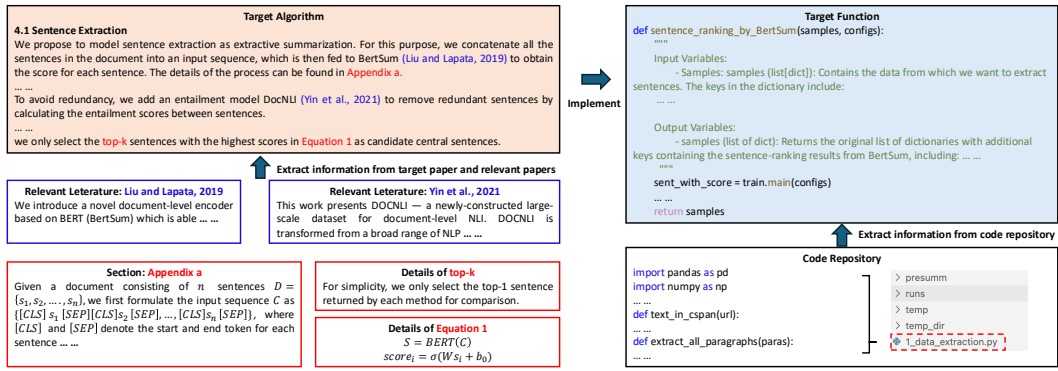

Figure A2: The task consists of two steps: *Algorithm Understanding* and *Code Implementation*. (Left) The model must extract an algorithm's workflow and details from the research paper, including descriptions and variable values from cited papers and other paper sections. (Right) Using this extracted information, the model implements the corresponding function in the code repository, correctly handling dependencies and API calls.

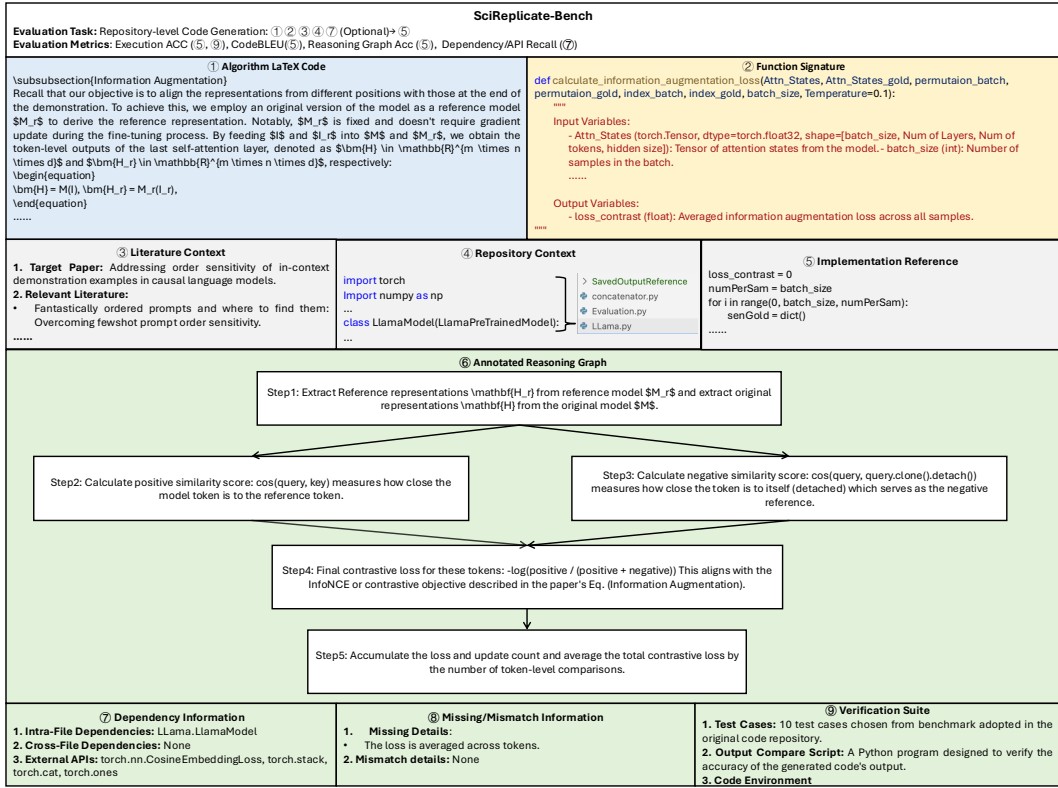

Figure A3: The overview of the SciReplicate-Bench.

Task: Match each generated step with its functionally equivalent reference step(s).

Input:
1. Target Method: Scientific research method described in LaTeX.
2. Reference Steps: Ordered steps from the original implementation, labeled [Ref 1], [Ref 2], etc.
3. Generated Steps: Ordered steps from an LLM-generated implementation, labeled [Gen 1], [Gen 2], etc.

Output Requirements:
1. For each generated step, identify the reference step(s) that implement the same specific functionality.
2. Format your answer as follows:
Gen 1: Ref X
Gen 2: Ref Y, Ref Z
Gen 3: -1
3. Matching criteria:
- Match based on functional equivalence, not textual similarity
- Steps must perform the same specific operation
- Steps must serve the same role in the overall algorithm
- Steps must produce equivalent results given the same inputs
4. Consider sequential position:
- Earlier generated steps likely match earlier reference steps
- Later generated steps likely match later reference steps
5. If a generated step has no clear equivalent or is ambiguous, output "-1"

6. Important:
- Ensure all reference indices actually exist in the reference steps
- Do not include explanations in your output
- Provide answers for all generated steps

Input Format:

[Target Method]
{target_method}

[Reference Steps]
{ref_comments}

[Generated Steps]
{gen_comment}

Your answer:

Figure A4: The prompt for node matching.

| Title | Conference |
|---|---|
| 1. From Zero to Hero: Cold-Start Anomaly Detection (Reiss et al., 2024) | Findings of ACL 2024 |
| 2. Addressing Order Sensitivity of In-Context Demonstration Examples in Causal Language Models (Xiang et al., 2024) | Findings of ACL 2024 |
| 3. Breaking the Ceiling of the LLM Community by Treating Token Generation as a Classification for Ensembling (Yu et al., 2024) | Findings of EMNLP 2024 |
| 4. Simple but Effective Compound Geometric Operations for Temporal Knowledge Graph Completion (Ying et al., 2024) | ACL 2024 |
| 5. When is Tree Search Useful for LLM Planning? It Depends on the Discriminator (Chen et al., 2024d) | ACL 2024 |
| 6. Functional Overlap Reranking for Neural Code Generation (To et al., 2023) | Findings of ACL 2024 |
| 7. Beyond Single-Event Extraction: Towards Efficient Document-Level Multi-Event Argument Extraction (Liu et al., 2024b) | Findings of ACL 2024 |
| 8. Unifying Dual-Space Embedding for Entity Alignment via Contrastive Learning (Wang et al., 2024a) | COLING 2025 |
| 9. Exploring Concept Depth: How Large Language Models Acquire Knowledge and Concepts at Different Layers? (Jin et al., 2024) | COLING 2025 |
| 10. TRANSMI: A Framework to Create Strong Baselines from Multilingual Pretrained Language Models for Transliterated Data (Liu et al., 2024c) | COLING 2025 |
| 11. Enhancing Knowledge Distillation of Large Language Models through Efficient Multi-Modal Distribution Alignment (Peng & Zhang, 2024) | COLING 2025 |
| 12. Document-level Claim Extraction and Decontextualisation for Fact-Checking (Deng et al., 2024) | ACL 2024 |
| 13. IRCAN: Mitigating Knowledge Conflicts in LLM Generation via Identifying and Reweighting Context-Aware Neurons (Shi et al., 2024) | Neurips 2024 |
| 14. RouterDC: Query-Based Router by Dual Contrastive Learning for Assembling Large Language Models (Chen et al., 2024b) | Neurips 2024 |
| 15. Unsupervised Homography Estimation on Multimodal Image Pair via Alternating Optimization (Song et al., 2024) | Neurips 2024 |
| 16. RAPTOR: Recursive Abstractive Processing for Tree-Organized Retrieval (Sarthi et al., 2024) | ICLR 2024 |
| 17. Less is KEN: a Universal and Simple Non-Parametric Pruning Algorithm for Large Language Models (Mastromattei & Zanzotto, 2024) | Findings of ACL 2024 |
| 18. Adaptive Contrastive Search: Uncertainty-Guided Decoding for Open-Ended Text Generation (Arias et al., 2024) | Findings of EMNLP 2024 |
| 19. MiniCheck: Efficient Fact-Checking of LLMs on Grounding Documents (Tang et al., 2024a) | EMNLP 2024 |
| 20. Nearest Neighbor Normalization Improves Multimodal Retrieval (Chowdhury et al., 2024) | EMNLP 2024 |
| 21. Neuron-Level Knowledge Attribution in Large Language Models (Yu & Ananiadou, 2023) | EMNLP 2024 |
| 22. RaTEScore: A Metric for Radiology Report Generation (Zhao et al., 2024) | EMNLP 2024 |
| 23. GREEN: Generative Radiology Report Evaluation and Error Notation (Ostmeier et al., 2024) | Findings of EMNLP 2024 |
| 24. Style-Specific Neurons for Steering LLMs in Text Style Transfer (Lai et al., 2024) | EMNLP 2024 |
| 25. Language-Specific Neurons: The Key to Multilingual Capabilities in Large Language Models (Tang et al., 2024b) | ACL 2024 |
| 26. Reasoning Paths Optimization: Learning to Reason and Explore From Diverse Paths (Chia et al., 2024) | Findings of EMNLP 2024 |
| 27. Lifelong Knowledge Editing for LLMs with Retrieval-Augmented Continuous Prompt Learning (Chen et al., 2024a) | EMNLP 2024 |
| 28. NeuroMax: Enhancing Neural Topic Modeling via Maximizing Mutual Information and Group Topic Regularization (Pham et al., 2024) | Findings of EMNLP 2024 |
| 29. Bridging Local Details and Global Context in Text-Attributed Graphs (Wang et al., 2024c) | EMNLP 2024 |
| 30. Advancing Adversarial Suffix Transfer Learning on Aligned Large Language Models (Liu et al., 2024a) | EMNLP 2024 |
| 31. SafeDecoding: Defending against Jailbreak Attacks via Safety-Aware Decoding (Xu et al., 2024) | ACL 2024 |
| 32. MaskLID: Code-Switching Language Identification through Iterative Masking (Kargaran et al., 2024) | ACL 2024 |
| 33. Towards Robust and Generalized Parameter-Efficient Fine-Tuning for Noisy Label Learning (Kim et al., 2024) | ACL 2024 |
| 34. Learning to Maximize Mutual Information for Chain-of-Thought Distillation (Chen et al., 2024c) | Findings of ACL 2024 |
| 35. GLiNER: Generalist Model for Named Entity Recognition using Bidirectional Transformer (Zaratiana et al., 2023) | NAACL 2024 |
| 36. End-to-End Beam Retrieval for Multi-Hop Question Answering (Zhang et al., 2023a) | NAACL 2024 |

Table A1: List of papers used in our Sci-Replicate benchmark.

Table A2: Reasoning Graph Accuracy of Sci-Reproducer Models Evaluated by LLMs and Human Annotators

| Sci-Reproducer Model | Evaluation Method | |
|---|---|---|
| | GPT-4o | Human (Mean ± Std) |
| GPT-4o-mini | 0.782 | 0.768 (0.066) |
| O3-mini-medium | 0.790 | 0.789 (0.033) |

Table A3: Pearson Correlation between LLM and Human Judgments of Reasoning Graph Accuracy

| Number of Data Points | r | P-value |
|---|---|---|
| 20 | 0.7518 | <0.005 |

| Approach | Error Ratio ($\downarrow$) |
|---|---|
| No Agent | 80.3 |
| Paper Agent | 83.3 |
| Code Agent | 29.4 |
| Sci-Reproducer | **24.9** |

Table A4: Syntax error across different settings.

| Categories | Examples |
|---|---|
| Hyperparameters and configurations | Thresholds, batch sizes, maximum iteration counts, exact numbers of clusters, initialization methods for variables or vectors, types of regularization (such as L1 or L2), and specific distance metrics (e.g., using L2 norm for Euclidean distances) |
| Numerical stability techniques | clamping values to avoid numerical instability, adding small constants during logarithmic calculations, managing division by zero scenarios, and addressing rounding and precision issues. |
| Implementation logic | Data splitting, application of dropout, formatting of input sequences, and handling special or edge cases in the input data. |
| Coding strategy | Caching for performance enhancement, retry mechanisms to handle failures, early stopping criteria, and strategies for memory optimization. |

Table A5: Some examples for different missing information categories.

Please complete the target function (or method) and provide the pure code output without any additional text. Include comments in the code following these specific guidelines:

Code Comments:
1. Focus on Reasoning: Comments should explain the reasoning behind the code generation process, as derived from the LaTeX description.
2. No Implementation Details: Avoid including any code-specific implementation details in the comments.
3. Mapping to LaTeX: Each comment must indicate which functionality described in the LaTeX code is implemented in the subsequent Python code snippet.
4. No overlap: The code snippets corresponding to each comment must not overlap, and nesting is not allowed.
5. Single Function: Implement the code within a single function (or method), without breaking it into multiple functions.
6. Import Package: Import all packages within the function (or method) to ensure the code is self-contained.
7. Format, the comment for each snippet should be in the following format:
# ------------------------------------------------------------------------
# Snippet x: Comment here
# ------------------------------------------------------------------------
# [Begin Snippet X]
Code snippet here
# [End Snippet X]

[Example]:
```python
def apply_token_level_transformation(
        token_representation,
        auxiliary_representation,
        transformation_indices,
        batch_mask,
        scaling_factor=0.01
):
        import torch
        import numpy as np
        transformed_tokens = token_representation.clone()

        for i, (start_idx, end_idx) in enumerate(transformation_indices):
            # ------------------------------------------------------------------
            # Snippet 1: We first verify if the current batch item is valid by checking the
            # batch_mask, analogous to referencing an in-context example $\mathbf{S}$
            # that has a corresponding reference $\mathbf{H_r}$ in the LaTeX snippet.
            # ------------------------------------------------------------------
            # [Begin Snippet 1]
            if batch_mask[i]:
            # [End Snippet 1]
                    # ------------------------------------------------------------
                    # Snippet 2: Here, we apply a basic shift to the token_representation by
                    # incorporating a slice from the auxiliary_representation, akin to
                    # combining $\mathbf{H}$ with a portion of $\mathbf{H_r}$ for
                    # enhanced alignment.
                    # ------------------------------------------------------------
                    # [Begin Snippet 2]
                    segment_main = transformed_tokens[i, start_idx:end_idx, :]
                    segment_aux = auxiliary_representation[i, start_idx:end_idx, :]
                    # [End Snippet 2]

                    # ------------------------------------------------------------
                    # Snippet 3: The transformation is a simple element-wise operation combined with
                    # the scaling_factor, illustrating a simplified version of the
                    # alignment concept from the LaTeX, which might involve more complex
                    # contrastive or attentional calculations.
                    # ------------------------------------------------------------
                    # [Begin Snippet 3]
                    updated_segment = torch.add(segment_main, torch.mul(segment_aux, scaling_factor))
                    transformed_tokens[i, start_idx:end_idx, :] = updated_segment
                    # [End Snippet 3]
                    # ------------------------------------------------------------------
            # Snippet 4: The final transformed_tokens are now partially aligned with the auxiliary
            # reference, reflecting the notion of augmenting token-level outputs
            # (Equation references in the LaTeX snippet would correspond to eq. (2-3) or
            # similar definitions of reference alignment).
            # ------------------------------------------------------------------------
            # [Begin Snippet 4]
            return transformed_tokens
            # [End Snippet 4]
```

Your answer:

Figure A5: The prompt for code generation.

[Task Overview]
Reproduce Python code corresponding to a LaTeX-based methodology from a scientific paper. However, due to the paper's length, it cannot be fully ingested by a large language model at once. Therefore, the solution requires two main steps:
1. Information Retrieval (Your Current Task): Extract relevant details, insights, and supporting information from the academic paper's LaTeX description and related literature.
2. Code Reproduction (Subsequent Task): Implement the Python code based on the information gathered and the provided LaTeX.

[Your Specific Focus]
You are tasked exclusively with Step 1: Information Retrieval. You must gather and organize all necessary details that will later be used to implement the Python code.

[Input]
1. List of sections: The paper includes the following sections (titles are provided for reference):

{Section_String}

2. LaTeX Description: The LaTeX code for the corresponding subsection in the paper, describing the algorithm implemented by the target function.

{latex_code}

3. Tools: Tools that can be adopted to gather external information during the information retrieval process.
* SearchPaper[query]
Description: When a variable, concept, or any other element appears in the target section without its full definition or sufficient details, use this action to search for the complete information in the full paper.
Parameters:
  - 'query' (string): A query describing the information that needs to be located within the full paper.
Examples:
  - If the LaTeX contains: "We use the concept of $X\_i$ to define Y," then the action should be: SearchPaper["The definition of $X\_i$"]
  - If the LaTeX contains: "The function $f(x)$ is defined based on the properties of $\mathcal{G}$.", then the action should be: SearchPaper["The properties of $\mathcal{G}$"]

* SearchSection[x]
Description: If the target section references another section in the paper with the title x, extract the information from the referenced section and return SearchSection[x].
Parameters:
  - w'x' (string): The title of the referenced section.
Example:
  - Latex: "The full derivation of our loss function can be found in method Section .", Action: SearchSection["method"]

* SearchLiterature[key, query]
Description: If the target section cites another paper (\cite{label}) and you determine that some information needs to be retrieved from that paper, return SearchLiterature[label, query], where query is the specific information you need to look for in the referenced paper.
Parameters:
  - 'key' (string): The citation key of the referenced paper. In LaTeX, when citing a paper, we use \cite{x}, where x represents the citation key.
  - 'query' (string): The specific information to search for in the referenced paper.
Example: I
  - Latex: "We adopt the metric proposed in \cite{wang2025}". Action: SearchLiterature["wang2025", "The proposed metric in the paper"]
  - Latex: "The algorithm is based on the work of \cite{smith2018}". Action: SearchLiterature["smith2018", "The algorithm details in the paper"]
  - Latex: "The dataset is based on the study by \cite{jones2020}". Action: SearchLiterature["jones2020", "The dataset details in the paper"]
[Instruction]
In order to complete code reproduction, it is first necessary to understand the algorithm described in the LaTeX description. The tools "SearchPaper", "SearchSection" and "SearchLiterature" should be used to retrieve relevant information from the paper to help you understand the methodology proposed in the latex description. For example:
1. If the LaTeX Description lacks the definition of a variable, use "SearchPaper" tool to find its definition.
2. If the LaTeX Description references other sections of the paper, use "SearchSection" tool to retrieve those sections and supplement the missing details.
3. If the LaTeX Description cites methods from other papers, use "SearchLiterature" tool to extract relevant information from the referenced papers.
[Action]
1. Apply a tool defined above to gather external information.
2. If you have gathered all the necessary information, fully understood the LaTeX code, and are prepared to proceed to the Code Reproduction stage, the appropriate action is "Finish"

[Observation]
1. If the action is apply predefined tool, then the observation should be the return response of the tool.

[Response Template]
Thought: I think ...
Action: SearchPaper[query] or SearchSection[label] or SearchLiterature[key, query] or Finish
Observation: Outcome of the action.

[Your Answer]
Please start information extraction step by step, strictly adhering to the provided template for the response format.

Figure A6: The prompt for Paper Agent.

You are a code assistant tasked with reproducing a Python function corresponding to a algorithm in the methods part of a scientific paper. The local coding environment includes a GPU and supports CUDA. I will provide the following information:

1. Repository structure: The organization of files within the code repository. This is a repository-level code generation task, so you should explore the repo thoroughly to extract useful code.
2. Target function: The definition of the python function you need to implement.
3. LaTeX description: The LaTeX code for the corresponding algorithm in the paper, describing the algorithm implemented by the target function.
4. The extracted information: The information extracted from the target paper, and relevant literature that can provide you more details when implement the target function.
5. Tools: Tools that can be adopted to gather external information during the generation process.

[Repository Structure]

{organization}

[Target Function]

The target function is located at "{Python_File_Path}". Its definition consists of the following components:

1. Input Variables
2. Output Variables

The definition is as follows:

{Target_Function}

[LaTeX Description]

{latex_code}

[Extracted Information]

The information is extracted from the paper and relevant literature by a paper search agent, which consists of a series of information points. When you implement the target function, you should refer to the extracted information to understand the target algorithm. When information from "Relevant Literature" conflicts with the target paper, always prioritize the information from the target paper.

The extracted information is as follows:

{extracted_info}

[Tools]
1. SearchWeb[Query]
Description: Perform a query using the Google search engine to retrieve relevant information. You can use this tool to search for examples of API usage, API definitions, bug fixes, implementations of similar algorithms, and more.
Parameters: Query (string): The search query to retrieve relevant information.
Example: SearchWeb["How to implement a neural network in PyTorch"]

2. SearchFile[M]
Description: Retrieve the content of a Python file from the current repository.
Parameters: M (string): The name of the python file to search for in the current repository.
Example: SearchFile["model.py"]

3. SearchCodeItem[M]
Description: Fetch information about a specific code item in the repository, including global variables, functions, methods, or classes.
Parameters: M (string): The name of the code item to search for in the current repository.
Example: SearchCodeItem["Model"]
Instruction:
In order to complete this task, it is necessary to use tools to search the code repository for context that can help implement the target function. For example:
1. Use "SearchFile" to retrieve the content of a Python file from the repository.
2. Use "SearchCodeItem" to find details about a specific code item within the repository.
3. Use "SearchWeb" to retrieve information from the website.
To effectively tackle the code reproduction task, follow a structured process that alternates between Thought, Action, and Observation steps:

[Thought]
1. Analyze the current situation.
2. Identify missing information from code. As it is a repo-level code generation task, you need to explore the relvant functions, classes, in the code repository.
3. Plan the next steps to gather the required information.

[Action]
1. Apply a tool defined above to gather external information.
2. If you are ready to generate the code, then the action should be "GenerateCode".

[Observation]
1. If the action is apply predefined tool, then the observation should be the return response of the tool.
2. If the action is "GenerateCode", then the observation is the result returned by the interpreter after executing the generated code.

[Response Template]
Thought: I think ...
Action: SearchWeb[Query] or SearchFile[M] or SearchCodeItem[M] or GenerateCode
Observation: Outcome of the action.

[Implementation Guidelines]
1. Step-by-step analysis of the LaTeX algorithm alongside extracted information.
2. Comprehensive repository exploration using provided tools.
3. Clean and efficient code implementation strictly matching the LaTeX algorithm.
4. Adherence to the structured Thought, Action, Observation response format.

[Your Answer]

Figure A7: The prompt for Code Agent.

Information Point 1: Hidden Representation of Last Token
Overview: Explanation of the variable $\bm{H}[i,-1]$ used in the consistency enhancement algorithm.
Source: Target Paper
Extracted Information:
  * $\bm{H}$ represents the token-level outputs of the last self-attention layer of the model.
  * $\bm{H} \in \mathbb{R}^{m \times n \times d}$, where $m$ is the batch size, $n$ is the sequence length, and $d$ is the dimensionality of the hidden layer.
  * The notation $[i,-1]$ refers to accessing the hidden representation of the last token for the $i$-th input in the batch.

Information Point 2: LoRA Adaptation
Overview: Description of the LoRA adaptation technique used in the algorithm.
Source: Target Paper
Extracted Information:
  * LoRA adaptation keeps all model parameters fixed except for the trainable query $W_Q$ and value $W_V$ matrices within the self-attention layers.
  * This technique integrates information from reference representations into the LoRA parameters during fine-tuning.
  * The rank of LoRA is set to 8 for all LLMs.

Information Point 3: Consistency Enhancement Methodology
Overview: Methodology for enhancing consistency in model outputs across a batch.
Source: Target Paper
Extracted Information:
  * The approach enforces similarity among the hidden representations of the last token across the input batch $I$.
  * The classification head parameters remain fixed during training.
  * The consistency loss $\mathcal{L}_{\text{con}}$ is calculated as $\sum_{i=1}^{m} (1-\text{cos}(\bm{H}[i,-1], \bm{H_a}))$.
  * $\bm{H_a}$ is the average of the last token's hidden representations across the batch: $\bm{H_a}= \frac{\sum_{i=1}^{m}{\bm{H}[i,-1]}}{m}$.

Figure A8: An example of output report of the Paper Agent.

