# OpenReview forum: "SciReplicate-Bench: Benchmarking LLMs in Agent-driven Algorithmic Reproduction from Research Papers"
_colmweb.org/COLM/2025/Conference — COLM 2025_

### Official Review · Reviewer_saNg · 2025-05-03

**Rating:** 5
**Confidence:** 3
**Ethics Flag:** 1

**Summary:**

The paper introduces SciReplicate-Bench, a benchmark designed to evaluate large language models (LLMs) on the challenging task of reproducing algorithms from recent NLP research papers. The benchmark comprises 100 tasks curated from 36 papers published in 2024, explicitly focusing on translating algorithm descriptions written in LaTeX and scattered scientific text into executable code within corresponding open-source repositories.

**Reasons To Accept:**

1. SciReplicate-Bench is a benchmark focused on reproducing realistic, algorithmic code implementations from academic publications rather than simple or toy tasks.
2. The introduction of reasoning graph accuracy evaluates not only final code correctness but also the intermediate reasoning steps that reflect the model’s comprehension.

**Reasons To Reject:**

1.  Considering that there are already many works utilizing multi-agent systems for scientific paper code generation, the novelty of the method in this paper is limited. This paper should elaborate more on the differences from these works. Although the appendix provides some explanations, the unique contribution of this paper is not clear. Some relevant works are as follows:
CORE-Bench: Fostering the Credibility of Published Research Through a Computational Reproducibility Agent Benchmark
Paper2Code: Automating Code Generation from Scientific Papers in Machine Learning
Agent Laboratory: Using LLM Agents as Research Assistants
The AI Scientist: Towards Fully Automated Open-Ended Scientific Discovery
 https://github.com/HKUDS/AI-Researcher
2.  In the experiments, comparisons with other works utilizing multi-agent systems for generating code from papers should be added.

---

> ### Author Response · Authors · 2025-06-03
> **Official Comment 2 by Authors**
>
> > **Reasons To Reject#2:** In the experiments, comparisons with other works utilising multi-agent systems for generating code from papers should be added.
>
> **Answer of Reasons To Reject#2:**
>
> As noted in our paper and also acknowledged in Paper2Code (Section 4.2), this is a relatively new and underexplored problem, and **there are currently no existing baselines specifically designed for the task of generating code directly from academic papers, making direct comparisons challenging.**
>
> I adopted **Paper2Code** and **ChatDev [1]** as baseline methods and conducted experiments using the GPT-4o-mini and O3-mini models. We provide the LaTeX code of the target algorithm, the definition of the code and the structure of the code repo as input for both baselines.
> Paper2Code typically generates full codebases by first planning the structure and then creating individual files. However, to reproduce **specific target functions and methods**, we're only using its **single-file generation** component as our baseline. The experimental results are shown in Table 1.
>
> **Table 1: Comparison of Baseline and Our Method Across Multiple Metrics Using GPT-4o-mini and O3-mini-medium**
>
> | Approach               | Exe Acc | CodeBLEU | Reasoning Graph Acc | Intra-File | Cross-File |  API  |
> |:----------------------:|:-------:|:--------:|:------:|:----------:|:----------:|:-----:|
> |       |         |          |    **GPT-4o-mini**      |            |            |       |
> | ChatDev                | 0.020   | 0.232    | 0.629  | 0.047      | 0.000      | 0.243 |
> | Paper2Code             | 0.030   | 0.239    | 0.712  | 0.024      | 0.000      | 0.217 |
> | **Sci-Reproducer (Ours)** | **0.170** | **0.303** | **0.768** | **0.576**  | **0.364**  | **0.362** |
> |     |         |          |    **o3-mini-medium**     |            |            |       |
> | ChatDev                | 0.040   | 0.258    | 0.746  | 0.035     | 0.000      | 0.311 |
> | Paper2Code             | 0.060   | 0.265    | 0.782  | 0.035     | 0.000      | 0.340 |
> | **Sci-Reproducer (Ours)** | **0.240** | **0.283** | **0.799** | **0.341**  | **0.061**  | **0.362** |
>
> From Table 1, we observe that our proposed method, **Sci-Reproducer**, consistently outperforms the two baseline approaches across all evaluation metrics. Specifically:
>
> - **Execution Accuracy and Recall**:
>   Sci-Reproducer achieves significantly higher performance compared to the baselines. For example, under the GPT-4o-mini setting, ChatDev and Paper2Code obtain execution accuracies of 0.020 and 0.030, respectively, while Sci-Reproducer reaches 0.170. This notable improvement is largely due to Sci-Reproducer’s ability to utilise predefined tools to execute the generated code and search relevant code repositories that are absent in the two baseline methods.
>
> - **Reasoning Graph Accuracy**:
>   Sci-Reproducer also outperforms both baselines in reasoning graph accuracy. For instance, with GPT-4o-mini, ChatDev and Paper2Code achieve scores of 0.629 and 0.712, respectively, while Sci-Reproducer achieves 0.768. This advantage can be attributed to the Paper Agent component in Sci-Reproducer, which leverages external tools to extract information from the paper and related literature.
>
> The experimental results further highlight the importance of the Execution Accuracy metric. **As the baseline methods lack the ability to execute the generated code using tools, their execution accuracy remains notably low.** In contrast, their CodeBLEU scores are relatively high, indicating that CodeBLEU is a less strict metric and may not fully capture functional correctness. **Therefore, incorporating Execution Accuracy provides a more precise and reliable assessment of the actual correctness of the generated code.**
>
> **Reference**
>
> [1] *ChatDev: Communicative Agents for Software Development*

---

> ### Author Response · Authors · 2025-06-03
> **Official Comment 1 by Authors**
>
> We appreciate the time and effort you've taken to review our work.
>
> > **Reasons To Reject#1:** This paper should highlight how SciReplicate-Bench differs from other approaches that use multi-agent systems for generating code from scientific papers.
>
> **Answer of Reasons To Reject#1:**
>
> To the best of our knowledge, our paper is **the first to introduce a benchmark specifically designed to evaluate LLMs’ ability to reproduce algorithmic code implementations from academic papers.** Our benchmark includes a carefully curated dataset with detailed annotations, standardised evaluation protocols, and a rigorous evaluation suite that checks the functional correctness of generated code via test cases and execution accuracy.
>
> Notably, **OpenAI’s PaperBench [1]** and **KAIST’s Paper2Code [2]** (**both released in April 2025, after our submission**) also propose benchmarks for paper replication, which shows growing interest in this direction and further highlights the relevance and timeliness of our contribution. Our work predates these releases, underscoring its novelty at the time of submission.
>
> The papers you mentioned are quite different from ours:
>
> - **CORE-Bench** focuses on **reproducing experimental results given existing code and data**.  In contrast, our benchmark targets a different and more challenging problem:  **generating code of the target algorithm from scratch based on paper content, without access to original implementations.**
>
> - **Agent Laboratory** and **The AI Scientist** aim to explore open-ended scientific discovery pipelines involving idea generation and paper writing.  While they include code generation components, **the code is typically written for LLM-generated ideas**, which are often less complex than real academic contributions.  Furthermore, **_these works do not rigorously assess the functional correctness of the generated code._**
>
> - **Paper2Code** shares a similar motivation but differs significantly in methodology.  **It uses LLMs as evaluators of code quality, which can be unreliable due to LLMs’ known limitations in judgment accuracy**.  Our benchmark adopts a more robust evaluation strategy based on executable test cases, ensuring more objective and reproducible assessments.
>
> We summarise the above works in **Table 1**.
>
> **Table 1: Comparisons of different benchmarks**
>
> ---
>
> | Benchmark         | Release date | Task                                                                                           | Execution based metric | LM based evaluation |
> |:-----------------:|:------------:|:----------------------------------------------------------------------------------------------:|:----------------------:|:-------------------:|
> | CORE-Bench        | 11/09/2024   | Given a code repository, Prepare Python environments, adapt code, and rerun experiments        | ✓                      | ✗                   |
> | PaperBench[1]     | 02/04/2025   | Generate the whole code repo for a scientific paper                                            | ✗                      | ✓                   |
> | Paper2Code[2]     | 24/04/2025   | Generate the whole code repo for a scientific paper                                            | ✗                      | ✓                   |
> | SciReplicate[ours]| 31/03/2025   | Generate the core function that corresponds to a core algorithm proposed in a scientific paper | ✓                      | ✓                   |
>
> ---
>
> **Reference**
>
> [1] *PaperBench: Evaluating AI's Ability to Replicate AI Research*
>
> [2] *Paper2Code: Automating Code Generation from Scientific Papers in Machine Learning*

---

> ### Author Response · Authors · 2025-06-07
> **Follow-up on Reviewer Feedback Before Rebuttal Deadline**
>
> We sincerely thank you for your time and thoughtful feedback. As the rebuttal period will conclude in a few days, we would greatly appreciate it if you could let us know whether our responses have sufficiently addressed your concerns, or if there are any remaining questions or points we could further clarify.

---

> ### Author Response · Authors · 2025-06-11
> **Follow-up 2 on Reviewer Feedback Before Rebuttal Deadline**
>
> We sincerely thank you once again for your time and valuable feedback. As today is the final day of the rebuttal period, we would greatly appreciate it if you could let us know whether our previous response has sufficiently addressed all concerns, or if there is anything further we can clarify before the deadline.

---

### Official Review · Reviewer_RNJE · 2025-05-08

**Rating:** 7
**Confidence:** 3
**Ethics Flag:** 1

**Summary:**

In this paper, the authors present SciReplicate-Bench, a benchmark that aims to model automated algorithm replication from scientific papers. Specifically, their benchmark is built 36 NLP papers from 2024 (the selection of date was intended to avoid leakage issues) and involves implementing specific functions from these papers (as such, it is a rather simplified version of the general replication task). As outlined at the beginning of Section 2, the benchmark provides the following input: 1) latex code for the function to implement; 2) a function signature for the target function; 3) the original paper for this function and; 4) details of the corresponding repository (this is some mismatch between the description of the input in lines 109-114 and the task description starting in line 131; I would encourage the authors to address this). All such information is extracted manually by annotators for the selected papers (like all such annotation tasks, this appears to be a timely process, taking 12 hours per paper).

To evaluate models on this benchmark, the authors propose 3 metrics (see lines 168-176): 1) execution-based accuracy (i.e., does the implemented function match the target output); 2) CodeBleu that measures code overlap with the reference code; 3) recall based on whether the code captures certain dependencies (the details here are less clear, I would encourage the authors to make the description more clear in 2.1.2). In addition, they also attempt to measure (what they call) “algorithm comprehension”, or the ability of the model to understand the target code, and do this by prompting models to document their code that facilitates the construction of a “reasoning graph” that they compare against a reference reasoning graph. This part of their evaluation is very interesting and  unique, though I have some questions about these graphs and whether the LLM-based evaluator the employ to compare graphs makes sense (see below).

They perform experiments using ReACT style agents. Their full approach (sci-reproducer) factorizes the problem into two agents, a paper agent and code agent, and they ablate each agent to measure it’s contribution (all results are in Table 3). They experiments with a comprehensive mixture of reasoning and non-reasoning models and find the tasks to be difficult according to all metrics (e.g., GPT-4o achieves only 0.170 execution accuracy). Additional analysis is performed to better understand certain error cases (e.g., logic errors and syntax errors) and differences in tool usage among different models (I didn’t find these results, summarized in Figure 2, to be particular well motivated or insightful; I would encourage the authors the consider using this space instead for some of the important details in the appendix, such as Figure A2).

**Questions To Authors:**

-- Was any manual evaluation performed on the node matching?
-- It's still unclear how precisely the recall was computed and how reliable this is.. Was this done via  some static analysis of the code? More details would be helpful here.

**Reasons To Accept:**

-- A new (expert annotated) benchmark for paper replication that I think others working in this area will benefit from (as I note below, however, I think this benchmark has some limitations compared with some missing related work that needs to be directly addressed).

-- A comprehensive set of baseline results that show the task to be difficult, along with a very strong multi-agent baseline (i.e., their "sci-reproducer").

-- Some novel ideas about code evaluation based on their "Code Reasoning Graph" that others working in this area might adopt to evaluating similar code generation problems.

**Reasons To Reject:**

-- Some key related work is missing. Specifically, a comparison with the SUPER benchmark (Bogin, et al 2024, EMNLP) (I think such a comparison should be included in Table 1). In some ways, the SUPER reproduction setup is more realistic than what's proposed in this paper (e.g., it focuses on building a full experiment pipeline rather than a single function). I would like to see this directly addressed by the authors.

-- (related to above) While SciReplicate-Bench aims to evaluate experiment replication, the specific way this operationalize this (i.e., as a single function implementation) addresses only a small and narrows aspect of this general problem.

-- Lack of human evaluation of automatic metrics. In particular, while the reasoning graph-based evaluation is interesting and novel, they do not appear to perform a manual evaluation of the automatic node matching and edge evaluation (from what I understand, such matching is performed by a LLM). Absent of this, it is very unclear how reliable these metrics are.

-- Similarly, it's unclear whether the CodeBleu, while a known metric is reliable. E.g., is there a strong correlation between code that executes correctly and CodeBleu?

---

> ### Author Response · Authors · 2025-06-03
> **Official Comment 3 by Authors**
>
> > **Reasons To Reject#4:** it's unclear whether the CodeBleu, while a known metric is reliable. E.g., is there a strong correlation between code that executes correctly and CodeBleu?
>
> **Answer of Reasons To Reject#4:**
>
> We calculated the Pearson correlation coefficient between CodeBLEU and Execution Acc based on different experimental setups and LLMs. By sampling 32 data points (CodeBLEU and corresponding Execution ACC), the correlation coefficient is detailed in Table 6:
>
> **Table 6: Statistical Comparison of CodeBLEU and Execution ACC: Variance, Deviation, and Correlation Coefficient.**
>
> |  N  | CodeBLEU variance | Execution Acc variance | Product of Deviation Scores |    r    | P-value |
> |:--:|:------------------:|:---------------------------:|:---------------------------:|:-------:|:-------:|
> | 32 |       0.045        |           0.307            |           0.105            | 0.8858  | <0.005  |
>
> As shown in Table 6, the correlation coefficient (r = 0.8858) indicates a very strong positive relationship between CodeBLEU and Execution Accuracy, with a high level of statistical significance (p < 0.005).
>
> Additionally, it is worth noting that Execution Accuracy exhibits a higher variance compared to CodeBLEU. This supports our decision to adopt Accuracy as the primary evaluation metric. However, the greater variance in Execution Accuracy also leads to some repeated low scores (as shown in Table 3 in our paper), which limits its ability to distinguish among cases where the generated code consistently fails to execute.
>
> To address this limitation, we require a metric that is both highly correlated with Execution Accuracy and capable of capturing fine-grained distinctions, especially when code execution fails. CodeBLEU serves this role well: **it maintains a strong correlation with Execution Accuracy (r = 0.8858) while exhibiting a lower variance (0.045)**, making it more effective in evaluating semantic differences in generated code.

---

> > ### Comment · Reviewer_RNJE · 2025-06-06
> >
> > Thank you for these additional details, they sufficiently clarify my questions and concerns. I would request that some of the correlational analysis and human evaluation you describe above be incorporated into the final paper. Due to this rebuttal, I am raising my score.

---

> > > ### Author Response · Authors · 2025-06-07
> > > **Thank You for the Feedback and Score Update**
> > >
> > > Thank you very much for your kind follow-up and for considering our clarifications. We truly appreciate your thoughtful suggestions and are grateful that you found our responses helpful. We will be sure to incorporate the correlational analysis and human evaluation into the final version of the paper, as per your recommendation.

---

> ### Author Response · Authors · 2025-06-03
> **Official Comment 2 by Authors**
>
> > **Questions To Authors#1:**  It's still unclear how precisely the recall was computed and how reliable this is.. Was this done via some static analysis of the code? More details would be helpful here.
>
> **Answer of Questions To Authors#1:**
>
> The calculation of Reasoning Graph Accuracy consists of two components: **node match** and **edge match**.
>
> - For **node match**, we use **LLM-as-a-judge** to determine which reference code comments correspond to each comment in the generated code.
> - For **edge match**, an automated approach is used. For each code segment corresponding to a comment, we extract the defined and used variables. If code segment A uses a variable defined in code segment B, we establish a directed edge from A to B (A → B), forming a directed graph. In the reference code, if there is a directed edge between two comments and both comments can be matched in the generated code, we then use a **BFS algorithm** to check whether a corresponding path exists between the matched nodes in the generated code. If such a path exists, the edge is considered a match.
>
> > **Reasons To Reject#3:** Lack of human evaluation of automatic metrics.
>
> **Answer of Reasons To Reject#3:**
>
> Thank you for your comments. To address your concern, we conduct human evaluation and apply more LLMs for evaluation.
>
> **1. Add human evaluation**
>
> we manually evaluated 10 tasks each from the outputs of the sci-reproducer models using GPT-4o-mini and O3-mini-medium. **This evaluation was carried out by three PhD students in computer science.**
>
> **Table 2: Reasoning Graph Accuracy of Sci-Reproducer Models Evaluated by LLMs and Human Annotators**
>
> ---
>
> | Sci-Reproducer Model (row) / Evaluation Model (column) | gpt-4o | Human Evaluation (Mean ± Std) |
> |:------------------------------------------------------:|:------:|:-----------------------------:|
> | gpt-4o-mini                                            | 0.782  | 0.768 (0.066)                 |
> | o3-mini-medium                                         | 0.790  | 0.789 (0.033)                 |
>
> ---
>
> **Table 3: Pearson correlation between LLM and human judgments of Reasoning Graph Accuracy**
>
> | Number of Data Points |    r    | P-value |
> |:-----------------:|:-------:|:-------:|
> |        20         | 0.7518  | <0.005  |
>
> As shown in Table 2, the average Reasoning Graph Accuracy across the 10 selected tasks demonstrates a strong alignment between human evaluations and LLM-based assessments for both GPT-4o-mini and O3-mini-medium.
>
> In addition, we compute the **Pearson correlation coefficient** between human evaluations and LLM-based assessments across all tasks, as shown in Table 3. The result (r = 0.7518) indicates a strong positive correlation between the two, further supporting the reliability of using LLMs for evaluation. Due to time constraints, we were only able to conduct a limited human evaluation during the rebuttal. A more comprehensive human evaluation will be included in a future version.
>
> **2. Apply more LLMs for evaluation**
>
> Recent replication benchmarks, including Paper2Code and PaperBench, have utilised LLMs as automated judges, with both studies identifying O3-mini-high as the most aligned with human judgment. In addition to GPT-4o, we further adopt O3-mini-high and GPT-4o-mini to evaluate reasoning graph accuracy, with the results presented in Table 4. We report the mean and standard deviation of the reasoning graph accuracy across three evaluation models, as shown in Table 5.
>
> **Table 4: Reasoning Graph Accuracy across different reproducer and evaluation model combinations**
>
> | Sci-Reproducer Model (row) / Evaluation Model (column) |  gpt-4o  | o3-mini-high | gpt-4o-mini |
> |:------------------------------------------:|:--------:|:------------:|:-----------:|
> | gpt-4o-mini                                |  0.768   |    0.767     |   0.766     |
> | gpt-4o                                     |  0.808   |    0.802     |   0.767     |
> | Claude-Sonnet-3.7                          |  0.794   |    0.828     |   0.794     |
> | Deepseek-v3                                |  0.778   |    0.797     |   0.767     |
> | o3-mini-medium                             |  0.799   |    0.797     |   0.794     |
>
> **Table 5: Mean and variance of Reasoning Graph Accuracy over three evaluation models**
>
> | Sci-Reproducer Model  |  Mean  | Standard Deviation |
> |:----------------------:|:------:|:------------------:|
> | gpt-4o-mini            | 0.767  |       0.001        |
> | gpt-4o                 | 0.792  |       0.022        |
> | Claude-Sonnet-3.7      | 0.805  |       0.019        |
> | Deepseek-v3            | 0.781  |       0.015        |
> | o3-mini-medium         | 0.797  |       0.002        |
>
> As shown in Table 5, **the Reason Graph Accuracy remains consistent across different evaluation models.** For Sci-Reproducer, the standard deviation of reasoning graph accuracy under different evaluation models is below 0.022, further demonstrating the reliability and stability of this metric.

---

> ### Author Response · Authors · 2025-06-03
> **Official Comment 1 by Authors**
>
> We appreciate the time and effort you've taken to review our work.
>
> > **Reasons to reject#1:**  A comparison with the SUPER benchmark (Bogin et al., 2024, EMNLP) is missing.
>
> **Answer of Reasons to reject#1:**
>
> Thank you for pointing out the SUPER benchmark. We will add it to Table 1. Key differences between SUPER and SciReplicate-Bench:
>
> - **SUPER supplies the full, original repository** (including the target algorithm) and asks the model to **prepare Python environments, adapt code, and rerun experiments**.
> - SciReplicate-Bench requires the model to **read a research paper, understand its algorithm** (including related work, mathematical details, and design choices), **retrieve any needed context**, and then **write code** that faithfully reproduces that core algorithm.
>
> > **Reasons to reject#2:**  While SciReplicate-Bench targets experiment replication, its focus on single-function implementation captures only a narrow slice of the broader replication challenge.
>
> **Answer of Reasons to reject#2:**
>
> Full codebase generation is the most general setting for “real-world” reproduction, which contains many components (data loaders, training pipelines, etc.). However, attempting to reproduce an entire project at once leads to **error localisation** becoming extremely difficult when a test fails. **By narrowing the task to a single function or method, which is typically the heart of a new algorithm, we can apply execution-based metrics in a fully automated, reproducible way.**
>
> Notably, **OpenAI’s PaperBench [1]** and **KAIST’s Paper2Code [2] (both released in April 2025, after our submission)** also propose benchmarks for paper replication. Their proposed tasks attempt full-codebase reproduction. However, their evaluation frameworks depend heavily on manually defined criteria and **use LLMs to judge code correctness**, which is an approach that is **susceptible to inconsistencies and known reliability issues**.  To ensure objectivity and reproducibility, we employ Execution Accuracy, which is a more rigorous approach than execution-free metrics ([3–5]).
>
> Moreover, our experiments show that even generating just the core function remains challenging for current models. We see this task as a foundational step toward full-pipeline replication and will clarify this motivation and positioning more explicitly in the revised paper.
>
> We summarise the above works in **Table 1**.
>
> **Table 1: Comparisons of different benchmarks**
>
> ---
> | Benchmark         | Release date | Task                                                                                           | Execution based metric | LM based evaluation |
> |------------------|--------------|------------------------------------------------------------------------------------------------|------------------------|---------------------|
> | SUPER            | 11/09/2024   | Given a code repository, Prepare Python environments, adapt code, and rerun experiments        | ✓                      | ✗                   |
> | PaperBench[1]    | 02/04/2025   | Generate the whole code repo for a scientific paper.                                           | ✗                      | ✓                   |
> | Paper2Code[2]    | 24/04/2025   | Generate the whole code repo for a scientific paper.                                           | ✗                      | ✓                   |
> | SciReplicate[ours] | 31/03/2025 | Generate the core function that corresponds to a core algorithm proposed in a scientific paper. | ✓                      | ✓                   |
>
> ---
>
> **References:**
>
> [1] *PaperBench: Evaluating AI's Ability to Replicate AI Research*
> [2] *Paper2Code: Automating Code Generation from Scientific Papers in Machine Learning*
> [3] *Evaluating Large Language Models Trained on Code*
> [4] *Execution-Based Evaluation for Open-Domain Code Generation*
> [5] *RepoCoder: Repository-Level Code Completion Through Iterative Retrieval and Generation*

---

### Official Review · Reviewer_fdpN · 2025-05-13

**Rating:** 5
**Confidence:** 4
**Ethics Flag:** 1

**Summary:**

The paper presents a new benchmark, SciReplicate-Bench, for reproducing scientific papers, which contains 100 tasks from 36 papers.
The authors present a new evaluation metric: reasoning graph accuracy, which checks the correctness of the implicit reasoning process behind code generation.
The tasks are also evaluated based on execution accuracy.

**Reasons To Accept:**

1. The task of reproducing numbers in research papers is natural.
2. The paper experiments with a paper agent with 3 actions and a coding agent with 4 actions, where the actions are all widely used in coding practice.

**Reasons To Reject:**

1. The setting of the tasks is not natural enough. The authors still focus on the traditional function/class generation setting (such as in RepoEval (https://arxiv.org/abs/2303.12570) or ClassEval (https://arxiv.org/abs/2308.01861)), where the coding agents aim to implement a specific function or class method in a given repo. This setting is not natural in two ways: (1) the repo context is extracted from the whole repo, not the commit where the target class/function has not been created yet, and (2) in practice, to implement some feature, we typically need to modify multiple classes and functions, instead of a single one.

2. The test case generation part is very important but is not clearly introduced in the paper (even in the Appendix). Here are some potential concerns about test generation: (1) how do you deal with function/classes with `print` or `logging` information? (2) how many tasks require GPU for computation and what kind of GPU? (3) The authors try to eliminate randomness as follows: "Annotators fixed random seeds and replaced non-deterministic operations (e.g., unordered sets) with deterministic equivalents to ensure consistent outputs across runs". What if the model-generated code contains some randomness??

---

> ### Author Response · Authors · 2025-06-02
> **Official Comment 2 by Authors**
>
> > **Reasons to reject#2:**
> > Test case generation is a crucial component, but the paper lacks a clear explanation of it. Several concerns remain unaddressed: (1) How are functions or classes with print/logging handled? (2) How many tasks require GPU execution, and what kind of GPU is used? (3) While the authors aim to eliminate randomness by fixing seeds and replacing non-deterministic operations, what happens if the model-generated code itself includes randomness?
>
> **Answer of Reasons to reject#2:**
>
> **(1) Test case generation:**
> Test cases are constructed using a subset of the original benchmark from the corresponding paper’s code repository. A limited set of inputs is selected for each task to cover the main functionality of the algorithm and keep the runtime efficient. Each function is tested with 10 input-output pairs, which are sufficient to verify the correctness and stability of execution across runs. We will add details in the paper to make it clear.
>
> **(2) Print and Logging Statements:**
> Each task in our benchmark targets a **single function**, and the correctness is determined by purely comparing the returned output variables (e.g., lists, arrays, tensors) against ground-truth values. Any print and logging statements in the reference implementations are **removed during preprocessing**, as they are not essential for verifying the correctness of the algorithm.
>
> **(3) GPU Requirements:**
> There are 77 tasks that need a GPU to run, but they are relatively light in computation and can be handled by a **single A100-80G GPU** or a comparable device.
>
> **(4) Handling Randomness:**
> Generated code can sometimes exhibit randomness, which may affect the reliability of Execution Accuracy as an evaluation metric. To address this, **all tasks in Scireplicate-Bench are verified to be reproducible when random seeds are fixed**. Additionally, our evaluation does not rely solely on Execution Accuracy. **We also incorporate non-execution-based metrics such as CodeBLEU and Recall, which are not impacted by randomness, to assess code correctness more robustly.**

---

> ### Author Response · Authors · 2025-06-02
> **Official Comment 1 by Authors**
>
> We appreciate the time and effort you've taken to review our work.
>
> > **Reasons to reject#1:**
> > The task setup is not natural. Like RepoEval and ClassEval, it focuses on generating a single function or class method, but this is unrealistic because (1) the context comes from the full repo, not a relevant commit where the target code doesn't yet exist, and (2) real-world development often requires editing multiple functions or classes, not just one.
>
> **Answer of Reasons to reject#1:**
>
> **(1) Difference between Scireplicate-Bench and RepoEval/ClassEval.**
> While our task format resembles prior benchmarks such as RepoEval and ClassEval, the purpose and scope of SciReplicate-Bench are fundamentally different:
>
> - **Algorithm Replication from Academic Papers.**
>   SciReplicate-Bench requires the model to read a research paper, understand its algorithm (including related work, mathematical details, and design choices), retrieve any needed context, and then write code that faithfully reproduces that core algorithm. This demands deep “literature-to-code” reasoning, rather than simply filling in a hole in a codebase.
>
> - **Novelty**
>   We are the **first** to propose a benchmark for paper replication. Notably, **OpenAI’s PaperBench [1] and KAIST’s Paper2Code [2] (both released in April 2025, after our submission)** share similar goals, further underscoring the novelty and relevance of SciReplicate-Bench.
>
> **(2) It’s unnatural to extract the repository context from the entire repo before making a commit.**
> During code generation, the reference implementation of the target function/method from the code repository is always hidden, so it is not exposed to the LLM. Moreover, in real-world code reproduction, it is common for researchers to build upon existing baseline repositories by adding new functions rather than writing code entirely from scratch. Therefore, providing repository context when implementing the target code aligns well with realistic code reproduction scenarios.
>
> **(3) Focusing on a Single Function or Method vs. “Full” Repo Changes.**
> Full codebase generation is the most general setting for “real-world” reproduction, which contains many components (data loaders, training pipelines, etc.). However, attempting to reproduce an entire project at once leads to **error localisation becoming extremely difficult when a test fails**. **By narrowing the task to a single function or method, which is typically the heart of a new algorithm, we can apply execution-based metrics in a fully automated, reproducible way.**
>
> PaperBench and Paper2Code attempt full-codebase reproduction. However, their evaluation frameworks depend heavily on manually defined criteria and use LLMs to judge code correctness, which is an approach that is susceptible to inconsistencies and known reliability issues. To ensure objectivity and reproducibility, we employ Execution Accuracy, which is a more rigorous approach than execution-free metrics ([3–5]).
>
> Moreover, our experiments show that even generating just the core function remains challenging for current models. We see this task as a foundational step toward full-pipeline replication and will clarify this motivation in the paper. We will clarify this motivation and positioning more explicitly in the revised paper.
>
> We summarise the above works in Table 1.
>
> **Table 1: Comparisons of different benchmarks**
>
> ---
>
> | Benchmark         | Release date | Task                                                                 | Execution based metric | LLM based evaluation |
> |------------------|--------------|----------------------------------------------------------------------|------------------------|---------------------|
> | RepoEval         | 20/10/2023   | Code completion in the software engineering area.                   | ✗                      | ✗                   |
> | ClassEval        | 3/8/2023     | Class-level code generation in the software engineering area.       | ✓                      | ✗                   |
> | PaperBench[1]    | 02/04/2025   | Generate the whole code repo for a scientific paper.                | ✗                      | ✓                   |
> | Paper2Code[2]    | 24/04/2025   | Generate the whole code repo for a scientific paper.                | ✗                      | ✓                   |
> | SciReplicate[ours] | 31/03/2025 | Generate the core function that corresponds to a core algorithm proposed in a scientific paper. | ✓ | ✓ |
>
> ---
>
> **References:**
>
> [1] *PaperBench: Evaluating AI's Ability to Replicate AI Research*
> [2] *Paper2Code: Automating Code Generation from Scientific Papers in Machine Learning*
> [3] *Evaluating Large Language Models Trained on Code*
> [4] *Execution-Based Evaluation for Open-Domain Code Generation*
> [5] *RepoCoder: Repository-Level Code Completion Through Iterative Retrieval and Generation*

---

> ### Author Response · Authors · 2025-06-07
> **Follow-up on Reviewer Feedback Before Rebuttal Deadline**
>
> We sincerely thank you for your time and thoughtful feedback. As the rebuttal period will conclude in a few days, we would greatly appreciate it if you could let us know whether our responses have sufficiently addressed your concerns, or if there are any remaining questions or points we could further clarify.

---

> ### Author Response · Authors · 2025-06-11
> **Follow-up 2 on Reviewer Feedback Before Rebuttal Deadline**
>
> We sincerely thank you once again for your time and valuable feedback. As today is the final day of the rebuttal period, we would greatly appreciate it if you could let us know whether our previous response has sufficiently addressed all concerns, or if there is anything further we can clarify before the deadline.

---

### Decision · Program_Chairs · 2025-07-08

**Decision:**

Accept

**Comment:**

This paper introduces SciReplicate-Bench, a novel and highly significant benchmark designed to evaluate Large Language Models (LLMs) in reproducing algorithms from NLP research papers into executable code, a task currently challenging for even state-of-the-art models like GPT-4o. The benchmark is praised for its high quality, clarity, and originality, notably introducing "reasoning graph accuracy" as a new metric alongside execution accuracy. This rigorous, well-structured work fills a critical gap in LLM evaluation, providing a realistic "literature-to-code" scenario that will significantly benefit research in automated algorithm replication and LLM capabilities, despite its current focus on single-function implementation.